# Biomarkers for prediction of neurological complications after acute Stanford type A aortic dissection: A systematic review and meta-analysis

Yi Si[1,2], Weixun Duan[3], Jiangang Xie[1], Chujun Duan[1], Shanshou Liu[1], Qianmei Wang[1], Xiaojun Zhao[1,2], Dan Wu[1], Yifan Wang[1], Lingxiao Wang[1], Junjie Li[1] *

1 Department of Emergency, Xijing Hospital, Air Force Medical University, Xi'an, China, 2 Department of Emergency, Affiliated Hospital of Yan'an University, Yan'an University, Yan'an, China, 3 Department of Cardiovascular Surgery, Xijing Hospital, Air Force Medical University, Xi'an, China

* drjunjieli@126.com

## Abstract

### Background

The predictive value of biomarkers such as neuron specific enolase (NSE), S100B, neurofilament (NFL), interleukin-6 (IL-6), coagulation factor R, and D-Dimer (DD) after acute Stanford A type aortic dissection (AAAD) with neurological complications has recently gained much attention from the research community. However, results from these studies are conflicting. This meta-analysis is conducted to assess the relationship between the biomarkers and the risk of neurological complications after AAAD.

### Methods

Two reviewers performed a systematic literature search across eight databases (CNKI, Wan Fang, VIP, CBM, PubMed, Web of Science, Cochrane Library, and EMBASE). The studies regarding biomarkers in AAAD patients published up to February 2022 were included. These studies were subjected to rigorous scrutiny and data extraction to determine the weighted mean difference (WMD) and the 95% confidence interval (CI), which were analyzed using the RevMan 5.4 and Stata software 14.0.

### Results

A total of 12 studies including 360 cases with neurological complications and 766 controls were incorporated into our meta-analysis. WMD analysis showed that there was a higher NSE levels in AAAD patients with postoperative neurological complications compared with controls (WMD = 0.640, 95% CI: 0.205 ~ 1.075, P = 0.004 < 0.005), and the level of S100B was related to the 6 h and 24 h postoperative neurological complications (6 h: WMD = 0.64, 95% CI: 0.27 ~ 1.02, P = 0.0007 < 0.001; 24 h: WMD = 0.281, 95% CI: 0.211 ~ 0.351, P < 0.001). Moreover, S100B levels at 6 hours after operation were significantly higher than that at 24 hours (WMD = 0.260, 95% CI: 0.166 ~ 0.354, P < 0.001).

**Data Availability Statement:** All relevant data are within the paper and its Supporting Information files.

**Funding:** The author(s) received no specific funding for this work

**Competing interests:** The authors have declared that no competing interests exist.

## Conclusion

NSE and S100B are both candidate biomarkers to predict postoperative neurological complications in patients with AAAD. Other markers are also valuable when used in conjunction with clinical judgement. The findings accentuate the necessity of further research to establish standardized values for these biomarkers in predicting neurological complications.

## Introduction

Acute Stanford A type aortic dissection (AAAD) is a life threatening disease, and surgery is the gold standard treatment. Given the high mortality, it would reach up to 90% in the first 24 hours if left untreated [1, 2]. Total aortic arch replacement is the first choice for saving the patients' lives in China [3], but the incidence of postoperative neurological impairment remains high around 10%-30% [1]. Even with brain protection technologies, such as hypothermia, cerebral perfusion, drug protection, and blood gas management, the incidence of postoperative neurological injury remains unchanged. Since the mid-1970s, the neurological complications related to aortic arch surgery have been divided into permanent neurological dysfunction (PND) and temporary neurological dysfunction (TPD), with the average incidence rate at 7.3% ~ 12.8% and 8.0% ~ 10.3%, respectively. PND is defined as a focal (embolic stroke) or global (diffuse coma) defect, characterized by sensory or motor disorders or lateralization of focal epileptic activity with corresponding imaging defects and persistent at discharge. TND is defined as a short-term clinical manifestation of short-term global dysfunction or neurocognitive decline (such as confusion, agitation, delirium, coma or myoclonic movement, no localized cerebral neurological symptoms, etc.) and is associated with long-term functional defects and decreased quality of life of patients. In addition, spinal cord nerve deficiency is also an important neurological complication, which seriously affects the patients' life quality and is life-threatening, bringing huge financial and mental burdens to their families. Therefore, it is of great significance to find out biomarkers for early prediction of postoperative neurological prognosis of AAAD patients. Diagnosis and prediction of postoperative neurological injury can be difficult because high clinical suspicion is required. Inflammatory markers including NSE and S100B have been researched intensively. S100B is an acidic calcium-binding protein isolated by Moore from the bovine brain in 1965. It is mainly distributed in glia and Schwann cells of the central nervous system and peripheral nervous system. When the cells in central nervous system are injured, S100B seeps from the cytosol into the cerebrospinal fluid, and then enters the blood through the damaged blood-brain barrier [4]. NSE is one of the enolases involved in glycolysis pathway. It is a dimer protein found in neural and neuroendocrine tissues. It is widely used as a tumor marker of neuroblastoma and small cell lung cancer. Like S100B, serum NSE contents are very low under normal conditions. When the central nervous system is injured, a large number of NSE is released from the cells and then enters the blood circulation [5]. Therefore, Previous studies have suggested that serum NSE and S100B are potential candidate markers for predicting neurological complications in patients with AAAD [5–7]. However, a recent study by Zhang k et al. [8] suggested that S100β and NSE levels showed no significant difference at 8-time points within 3 days after the operation. Here, we aimed to explore the biomarkers related to the prognosis of neurological function after AAAD through meta-analysis to provide the evidence-based basis for early clinical prediction and management.

Furthermore, other markers including NFL, IL-6, DD and coagulation factor R were expected to be predictors in some studies. These markers were not investigated sufficiently to

meet the standard of meta, thus this study will conduct a systematic review of published studies related to them and evaluate the utility of these clinical investigations.

## Methods

We conducted this systematic review and meta-analysis in accordance with the Preferred Reporting Items for Systematic Reviews and Meta-Analysis (PRISMA) guidelines. The review was registered on the PROSPERO database (PROSPERO ID = CRD4202201770).

### Search strategy

A systematic search of publications listed in the databases (CNKI, Wan Fang, VIP, CBM, PubMed, Web of Science, Cochrane Library and EMBASE) from commencements to February 2022 was conducted using the following search terms: ("aortic dissection" or "acute type A aortic dissection" or "DeBakey type I aortic dissection" or "thoracoabdominal aortic aneurysm" or "cardiopulmonary bypass brain injury" or "acute type A aortic dissection" or "open surgical repair" or "aortic arch replacement") and ("biomarker *" or "neurofilament protein L" or "NSE" or "neuron specific enolase" or "S100B" or "interleukin-6" or "IL-6" or "D-dimer"). All references in the retrieved articles were also scanned to identify other potentially available studies.

### Inclusion and exclusion criteria

The inclusion criteria encompassed the following: (1) cohort study or case-control study; (2) diagnosis of AAAD or related cardiac surgery by CT / MRI, echocardiography, and blood gas analysis, etc.; (3) the definition of neurological complications must be clearly; (4) the mean and standard deviation (SD) of serum biomarkers can be obtained or calculated directly; (5) published by Chinese or English. Studies were excluded based on the following criteria: (1) not a cohort study or case-control study (including animal experiment, in vitro experiment, pharmacoeconomic research, and drug metabolism research); (2) overlapping samples of the same research center; (3) incomplete data information; (4) other types of aortic dissection; (5) the article was a case report, meta-analysis, review, or abstract from the conference.

### Data extraction

Based on the inclusion and exclusion standards, two researchers independently reviewed the full text of all included studies and extracted the following relevant information: first author, publication time, region, sample size, biomarkers levels, type of operation, judgment standard, and time of neurological complications prognosis, time of blood draw, specificity, sensitivity, cut-off value, and other observation indexes. The differences between the data extracted by the two researchers were reviewed by the third researcher.

### Quality assessment

We used the Newcastle-Ottawa Scale (NOS) standard to evaluate the quality of each study [9]. Three aspects were assessed: subject selection, inter-group comparability, and exposure factor measurement. The full NOS score is 9 points, with a score $\geq 6$ indicating high quality.

### Statistical analysis

RevMan software (version 5.4; Cochrane Collaboration, Oxford, UK) and Stata software (version 14.0; Stata Corporation, College Station, TX, USA) were used in our article for all data processing and analysis. The primary data extracted in this study were sample size, as well as

the mean, median, interquartile range (IQR), and SD of biomarkers concentration. In this study, we extracted the median to represent the mean, and converted the IQR into SD value through Mean Variance Estimation online software. Due to the same measurement units of the concentration of the biomarker in the included studies, the weighted mean difference (WMD) and 95% confidence interval (CI) were selected as the effective size (ES) for statistical analysis for continuous outcomes. A forest plot was used to show the characteristics of the results of various studies. P < 0.05 indicated a statistically significant difference. We used the Q test and $I^2$ statistics to analyze the heterogeneity of the included studies. The fixed-effect model was employed for data combination if P $\geq$ 0.05 and $I^2 \leq$ 50%, indicating no statistical heterogeneity between the studies. The random-effect model was adopted if the converse was true. To further investigate the potential influencing factors of heterogeneity, we conducted a subgroup analysis of the study based on surgery type. Furthermore, a sensitivity analysis was carried out to assess the stability of the results. Repeated data analyses were performed, with each done after removing one study at a time. In addition, we also performed a potential publication bias assessed by visual inspection of the funnel plot, an asymmetric plot suggested a possible publication bias would be corrected by the shear compensation method.

## Results

### Characteristics of eligible studies

A total of 5891 related records were initially retrieved through each database. After screening the titles or abstracts, 5868 studies were excluded because of duplicate data, other types of publications (animal/reviews/editorial/comments/letter), or non-relevant research. The full-length papers of the remaining 23 studies were retrieved and assessed for eligibility. Of the retrieved studies, a total of 12 met our inclusion criteria and were included in our systematic review, 8 were incorporated into our meta-analysis. A PRISMA flow diagram can be seen in (Fig 1). Table 1 and S1 Table provide a summary of the key characteristics and risk assessment of publication bias of the included studies. Table 2 exhibits the cut-off values, sensitivity, specificity, area under the curve (AUC), and other observation indexes for the biomarkers in the 12 included studies.

### NSE

Five studies reported that NSE levels at 24 hours after surgery were associated with the increased risk of postoperative neurological complications (WMD = 7.14, 95% CI: 2.19 ~ 12.09, P = 0.005, $I^2$ = 98%) (Fig 2A). There was a marked heterogeneity in the data. We conducted a sensitivity analysis by excluding each study in turn for heterogeneity. We found that the marked heterogeneity came from the studies of Kimura, F and Zou Yi xi which were not exact AAAD (Table 1). In our subgroup analyses according to the different types of surgery, an increased risk of postoperative neurological complications should be noted compared with controls (WMD = 0.75, 95%CI: 0.24~1.25, P = 0.004<0.005, $I^2$ = 0%) (Fig 2B) and analytic heterogeneity were very low. Additionally, the funnel plot seemed to be asymmetrical and we corrected it with the shear compensation method (S1A and S1C Fig). After correction with two virtual studies and iterated for three times, the results showed no much difference from that before clipping (WMD = 0.640, 95% CI: 0.205 ~ 1.075, P = 0.004 < 0.005) (S1B Fig). It can be considered that the existing meta-analysis results are relatively stable.

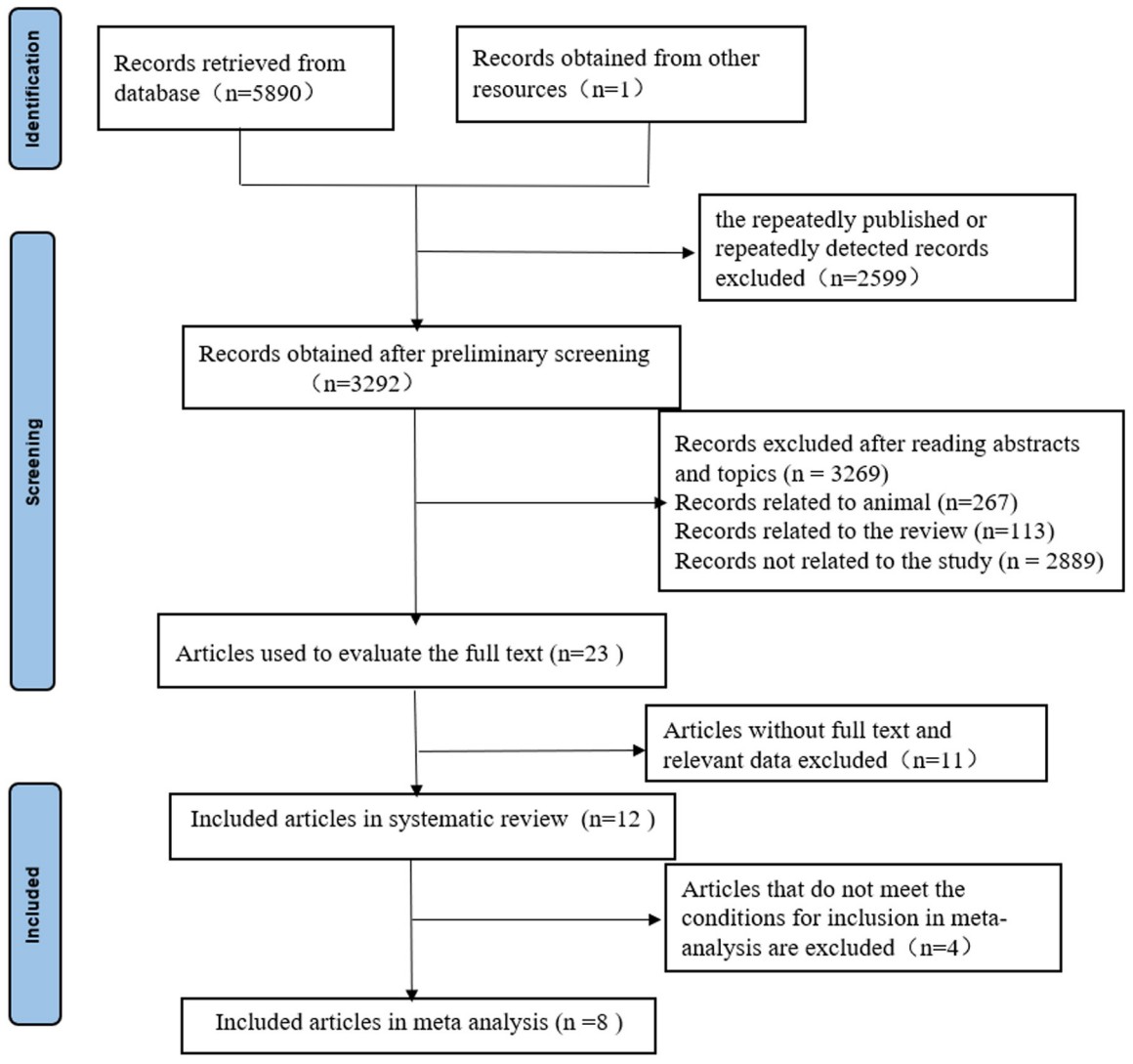

**Fig 1. PRISMA flow diagram depicting study selection algorithm.**

## S100B

Five studies reported that S100B levels at 6 hours after surgery were associated with the increased risk of postoperative neurological complications (WMD = 0.64, 95% CI: 0.27 ~ 1.02, P = 0.0007< 0.001, $I^2$ = 97%) (Fig 3A). There was a statistical heterogeneity among the studies. We also conducted a sensitivity analysis by excluding each study in turn for heterogeneity. There were no significant changes (S2A Fig). Additionally, the funnel plot seemed to be asymmetrical and we corrected it with the shear compensation method (S2B and S2D Fig). After correction by iterating twice, the results remain unchanged (S2C Fig) which indicated that the results were stable.

Six studies reported that S100B levels at 24 hours after surgery were associated with the increased risk of postoperative neurological complications (WMD = 0.26, 95% CI: 0.12 ~ 0.39, P = 0.0002 < 0.001, $I^2$ = 94%) (Fig 3B). There was a marked heterogeneity in the data. We conducted a sensitivity analysis by excluding each study in turn for heterogeneity. We found that

**Table 1. Characteristics of all studies included in the review.**

| First author | Year | Country | Type of surgery | Sample size (n) | | Design | NOS Score |
|---|---|---|---|---|---|---|---|
| | | | | with neurological complications | Without neurological complications | | |
| Lang Q [10] | 2021 | China | AAAD | 12 | 39 | Retrospective | 6 |
| Zhang K [8] | 2021 | China | AAAD | 15 | 73 | cohort study | 7 |
| Wan Z [6] | 2021 | China | AAAD | 15 | 25 | cohort study | 7 |
| Lv X [11] | 2021 | China | AAAD | 31 | 190 | Retrospective | 8 |
| Peng X [5] | 2020 | China | AAAD | 21 | 76 | cohort study | 6 |
| Kimura F [7] | 2020 | Japan | AAAD Thoracic aortic aneurysm | 12 | 48 | Retrospective | 7 |
| Fang M [12] | 2015 | China | AAAD | 169 | 166 | Retrospective | 7 |
| Zhang Z [13] | 2012 | China | CPB | 22 | 10 | Retrospective | 7 |
| Dalyanoglu H [14] | 2012 | Germany | AAAD | 14 | 33 | Retrospective | 7 |
| Zou Y [15] | 2011 | China | Thoracic abdominal aortic dissection | 4 | 26 | cohort study | 8 |
| Chen B [4] | 2009 | China | CPB | 30 | 32 | cohort study | 7 |
| Liu X [16] | 2007 | China | CPB | 15 | 48 | cohort study | 6 |

the marked heterogeneity came from the studies of WAN, Z, Peng Xiao le and Zou Yi xi which deviated greatly (S3A Fig). When the above studies were removed, the heterogeneity was significantly reduced (WMD = 0.33, 95% CI: 0.27 ~ 0.39, P < 0.0001, $I^2$ = 46%) (Fig 3C). Additionally, the funnel plot was asymmetric and we corrected it with the shear compensation method (S3B and S3D Fig). After correction with two virtual studies and iterated four times, no difference was observed from that before clipping, indicating a stable result (WMD = 0.281, 95% CI: 0.231 ~ 0.331, P < 0.001) (S3C Fig).

Five studies reported that the level of S100B at 6h increased more significantly than that at 24h after surgery (WMD = 0.36, 95% CI: 0.07 ~ 0.65, P = 0.01 < 0.05, $I^2$ = 93%) (Fig 4A), There was a marked heterogeneity in the data. We conducted a sensitivity analysis by excluding each study in turn for heterogeneity. We found that the marked heterogeneity came from the studies of WAN, Z and Chen Bin which deviated greatly (S4A Fig). When the above studies were removed, the heterogeneity was significantly reduced (WMD = 0.29, 95% CI: 0.17 ~ 0.40, P < 0.00001, $I^2$ = 0%) (Fig 4B). Additionally, the funnel plot was asymmetric and we corrected it with the shear compensation method (S4B and S4D Fig). After correction with two virtual studies and iterated three times, the results could be considered to be stable (WMD = 0.260, 95% CI: 0.166 ~ 0.354, P < 0.001) (S4C Fig). Lastly, the consequence of these studies summarized in Table 3.

## Discussion

The incidence of neurologic injury is higher in AAAD than other procedures owing to the involvement of hemodynamic factors in the form of embolisms caused by surgical manipulation of cervical branches and reduction in systemic blood pressure due to CPB [7]. Therefore, early therapeutic intervention is essential to minimize the sequelae of these dangerous complications. This meta-analysis summarizes a total of 8 studies fit the inclusion criteria and confirmed that there was a statistically significant association between the biomarkers concentration and the risk of postoperative neurological complications. Both serum NSE and S100B were significantly increased in the patients with neurological complications compared with the control cohorts. The occurrence of postoperative neurological complications is caused

**Table 2. Cut-off value, sensitivity, specificity, AUC and other observation indexes for NSE, S100B, NFL, IL-6, DD and coagulation factor activity R.**

| Biomarker | First author | Neurological complications | | | Time of blood draw | Outcomes |
|---|---|---|---|---|---|---|
| | | Type | evaluation standard | Evaluation time | | |
| NSE | Wan, Z [6] | POCD | MMSE | Before surgery, the day after extubation, and the 7th day after surgery | T0、T2、T4、T7 | The sensitivity was 92%, the specificity was 67%, and the AUC was 0.77 (95%CI: 0.60~0.94) |
| | Peng X [5] | POCD | MMSE | Before and 7 days after surgery | T0、T1、T2、T6、T9 | NS |
| | Kimura,F [7] | PND TND | CT MRI | Before and 24 hours after surgery | T0、T9 | The PND cutoff was 43.56 ng/mL (sensitivity was 100%; specificity was 96.3%), OR is 0.61 |
| | Dalyanoglu, H [14] | PND TND | CT | NS | NS | NS |
| | Zou Y [15] | Spinal cord injury | NIHSS ISCOS | Before surgery, 72 hours after surgery and at discharge, follow-up time was 3 months | T1、T2、T3、T5、T7、T8、T9、T10 | The sensitivity was 100%, the specificity was 96.2%, and the cutoff value was 16.84 µg/L |
| S100B | Wan, Z [6] | POCD | MMSE | Before surgery, the day after extubation, and the 7th day after surgery | T0、T2、T4、T7 | The sensitivity was 48%, the specificity was 87%, and the AUC was 0.71 (95% CI, 0.55–0.87) |
| | Peng X [5] | POCD | MMSE | Before and 7 days after surgery | T0、T1、T2、T6、T9 | The OR of S-100B> 300 ng/L at 24 h after surgery was 10. 827 |
| | Zou Y [15] | Spinal cord injury | NIHSS ISCOS | Before surgery, 72 hours after surgery and at discharge, follow-up time was 3 months | T1、T2、T3、T5、T7、T8、T9、T10 | The sensitivity was 100%, the specificity was 96.2%, and the cutoff was 1.4 µg/L |
| | Zhang Z [13] | Brain Injury | NIHSS | Before and 2 days after surgery | T0、T3、T4、T7、T9 | It is more meaningful to evaluate the prognosis 6 hours after the end of CPB |
| | Chen B [4] | POCD | Newman | 1 day before surgery and 9 days after surgery | NS | NS |
| | Liu X [16] | POCD | Software testing scale | 1 day, 2 days, 7 days after surgery | T0、T3、T4、T8、T9 | NS |
| NFL | Zhang, K et al. [8] | PND | CT | postoperative | T0、T1、T3、T7、T8、T9、T10、T11 | The AUC of NFL at 12 h after surgery was 0.834 (95%CI, 0.723–0.951, P<0.001), the sensitivity was 86.7%, the specificity was 71.6% |
| IL-6 | Lv, X [11] | Delirium | RASS | For the first three days after surgery, assess twice a day in the intensive care unit (ICU) or general ward | T0、T9、T10、T11 | The 24h postoperative cut-off value was 266.80 pg/ml, and the sensitivity was 71.0% with a specificity of 69.8% |
| DD | Lang Q [10] | PND TND | CT | Day 1 to Day 11 after surgery | T0 | OR = 0.934 (95% CI: 0.846 ~ 1.031) |
| | Fang M [12] | Delirium | RASS CAM-ICU | postoperative | T0 | OR = 2.48 (95% CI: 1.347 ~ 4.564) |
| coagulation factor activity R | Lang Q [10] | PND TND | CT | Day 1 to Day 11 after surgery | T0 | OR = 2.013 (95%CI: 1.008~4.021) |

MMSE, mini-mental status examination; POCD, postoperative cognitive dysfunction; NIHSS, national institutes of health stroke scale; ISCOS, international neurological classification of spinal cord injury; CPB, cardiopulmonary bypass; OR, odds ratio; AUC, area under the curve; NS, not stated; T0, before the operation;T1, when circulation is stopped;T2, when rewarming to 36˚C;T3, within 1 hour after the operation;T4, 1 hour after the operation;T5,2 hours after the operation;T6,4 hours after the operation; T7,6 hours after the operation;T8,12 hours after the operation;T9, 24 hours after the operation;T10,48 hours after the operation; T11,72 hours after the operation; NOS, newcastle-ottawa scale

by the synergy of multiple factors and mechanisms, including endogenous factors related to patients and the disease itself, such as poor perfusion and involvement of vessels on the aorta, and also exogenous injuries from surgical strategy [17]. S100B is usually not detected in serum, increases only under stroke, subarachnoid hemorrhage, and cardiopulmonary bypass, and is thermostable. The serum concentration of S100B is not affected by heparin, protamine, propofol, and hemolysis, so the sample can be taken at any time during the operation, and the high

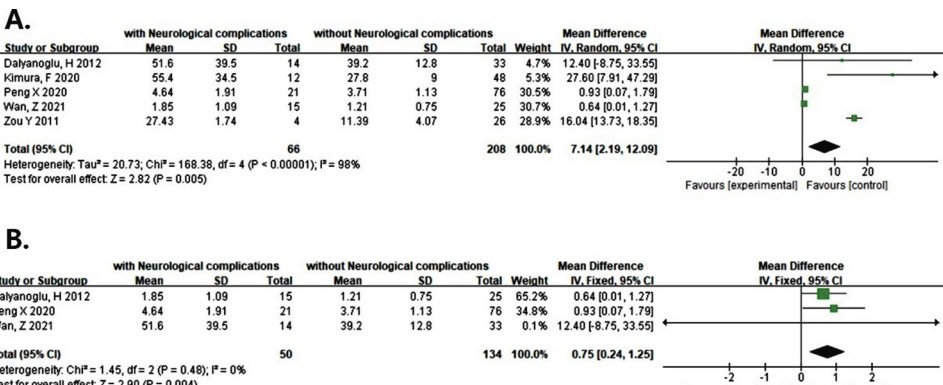

**Fig 2.** (A) Forest plot of NSE predicting neurological complications at 24 hours after operation. (B) Forest plot of the pooled result after corrected by the different types of surgery according to the sensitivity analysis.

concentration exists in glial cells and Schwann cells. Like S100B, NSE contents are very low under normal conditions. When the central nervous system is injured, a large number of NSE is released from the cells and then enters the blood circulation [5]. Therefore, S100B and NSE are biochemical indicators to predict neurological function after AAAD.

Meanwhile, most studies predicting neural function state are based on the S100B concentration before and after cardiopulmonary bypass (CPB), but the change in the CPB time and the CPB-induced neural function remains unclear. Zhang et al. [13] showed that S100B in patients was at the normal level before operation and increased rapidly after CPB, especially in the patients with moderate and severe brain injury. S100B in the asymptomatic group and mild brain injury group reached the peak at 1h after CPB and decreased at 6h after CPB, while plasma S100B in moderate and severe brain injury group continued to rise and reached the peak at 6h after CPB, and then S100B concentration gradually decreased, but still higher than the normal upper limit at 24h after CPB. The results suggest that the level and duration of

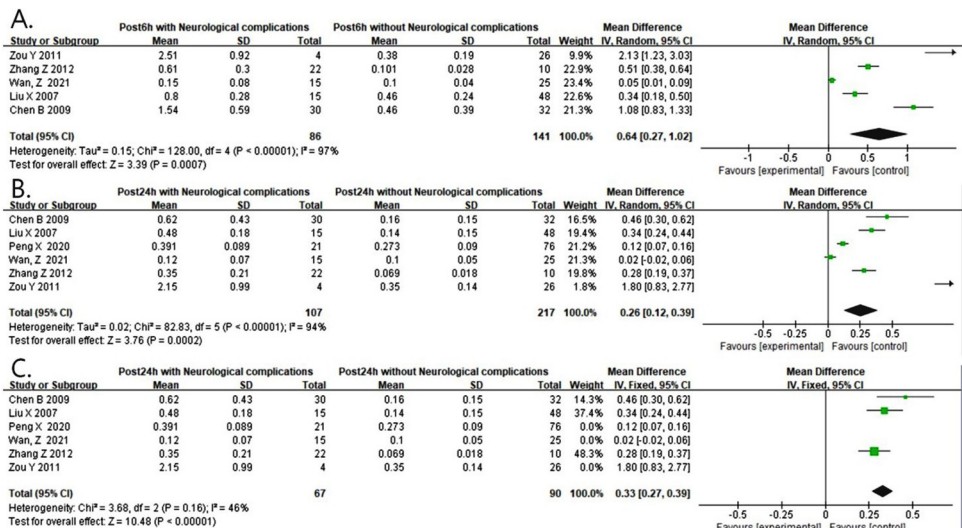

**Fig 3.** (A) Forest plot of S100B predicting neurological complications at 6 hours after operation. (B) Forest plot of S100B predicting neurological complications at 24 hours after operation. (C) Forest plot of the pooled result after corrected by the sensitivity analysis.

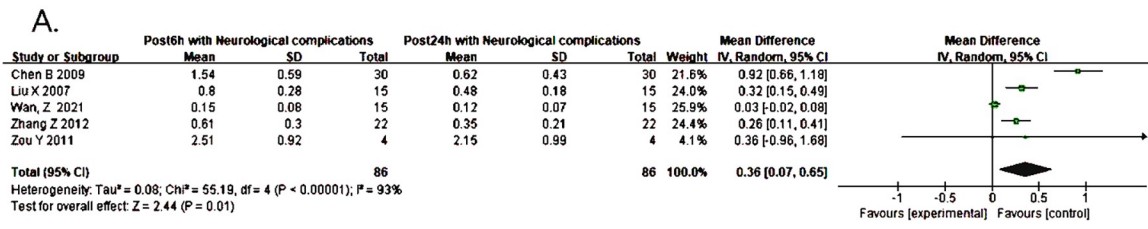

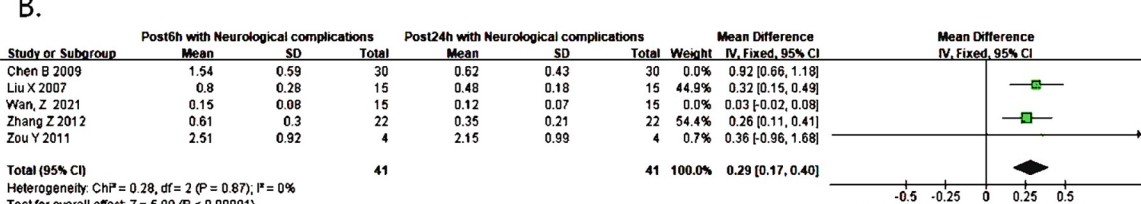

**Fig 4.** (A) Forest plot of S100B predicting neurological complications at 6 and 24 hours after operation. (B) Forest plot of the pooled result after corrected by the sensitivity analysis.

S100B are positively correlated with the severity of brain injury. In the current meta-analysis, we found that the level of S100B increased more significantly at 6h than at 24h after operation, which is consistent with the results from Zhang's study. Therefore, dynamic observation of patient S100B levels can not only determine the existence of brain injury and evaluate the injury degree after the operation, but also provide more suitable blood collection time for follow-up researchers.

In addition, this study also conducted a systematic review of other markers related to AAAD with postoperative neurological complications comprehensively and rigorously extracted the data about the time of blood draw, neurological complications evaluation standard and time, type and outcome to accentuate the necessity of further research to establish standardized values for these biomarkers in predicting neurological complications. A total of 12 studies fit the inclusion and some studies indicated that blood exposure to the external environment during CPB would activate the body's immune system, leading to perioperative systemic inflammatory response and the release of various pro-inflammatory cytokines, such as tumor necrosis factor α, interleukin (IL)-1β, IL-6, and IL-8, etc. The inflammatory factors further activate indoleamine 2,3-dioxygenase, the rate-limiting enzyme of the kynurenine pathway (KP), resulting in over-activation of central and peripheral KP, neurotoxicity, cognitive function, and even neurological complications [5]. LV et al. [11] found that plasma IL-6 is a potential biomarker for predicting postoperative delirium in patients with AAAD, with a

**Table 3. Summary of meta-analysis results.**

| Biomarker | Time | WMD(µg/L)(95%CI), p-value | $I^2$ | Subgroup based on Sensitivity analysis | $I^2$ | Analysis results after correction by trim and filling method |
|---|---|---|---|---|---|---|
| NSE | 24 hours after operation | 7.14(2.19–12.09)<0.01 | 98% | 0.75 (0.24–1.25) <0.005 | 0% | 0.640(0.250–1.075)< 0.005 |
| S100B | 6 hours after operation | 0.64(0.27–1.02)<0.001 | 97% | __ | __ | __ |
| | 24 hours after operation | 0.26(0.12–0.39)<0.001 | 94% | 0.33(0.27–0.39)<0.00001 | 46% | 0.281(0.231–0.331)< 0.001 |
| | 6 and 24 hours after operation | 0.36(0.07–0.65) <0.05 | 93% | 0.29(0.17–0.40)<0.00001 | 0% | 0.260(0.166–0.354)< 0.001 |

WMD, weighted mean difference;—, No significant change.

preoperative cut-off value of 95.45pg/ml, sensitivity of 61.3%, specificity of 79.4%, and AUC value of 0.73 (95% CI, 0.632 ~ 0.833), while the 24-hour post-operation cut-off value was 266.80pg/ml, sensitivity was 71.0% and specificity was 69.8% (Table 2). Besides, cerebral small vessel disease (CSVD) is one of the main causes of stroke, and neurofilament light chain protein (NFL) is the main component of neurofilament core, which is related to the occurrence of subcortical small infarcts and new CSVD. (NFL has the highest predictive value for neural function in patients with AAAD at 12h and 24h after operation, with AUC values of 0.834 (95% CI: 0.723 ~ 0.951, P < 0.001) and 0.748 (95% CI: 0.603 ~ 0.894, P = 0.004), respectively (Table 2).

Studies revealed that the incidence of cerebral nervous system complications increases, when the coagulation function is abnormal (hypercoagulable state) [18]. DD is a specific degradation product of cross-linked fibrin, representing the activation of the coagulation and fibrinolysis system. DD is directly related to disseminated intravascular coagulation (DIC). Lang et al. [10] took postoperative PND and TND of AAAD patients as the endpoints of a neurological complication study and found that the OR value of DD predicting neurological function before the operation was 0.934 (95% CI 0.846 ~ 1.031), while Fang et al. [12] took postoperative delirium as the endpoint of neurological complication study and concluded that the OR value of DD predicting neurological function before the operation was 2.48 (95% CI 1.347 ~ 4.564). In addition, the R value of coagulation factor activity indicates the time from putting into the blood sample to the formation of the first fibrin, which is also related to the patient's hypercoagulable state (the shortening of the R value indicates the increase of coagulation factor activity and the hypercoagulable state of the blood). Lang et al. [10] also showed in the same study that the OR value of coagulation factor activity R before the operation for predicting neurological function was 2.013 (95% CI 1.008 ~ 4.021).

Several limitations of this meta-analysis should be noted:(1) only 12 Chinese and English studies were included in the study, of which 10 studies were from China and the time duration was a little long, which may cause publication bias; (2) the types of operation and the definitions of neurological complications were not unified, these may partially result from the source of heterogeneity; (3) given that all included studies were observational, the possibility of residual confounding by unmeasured factors cannot be eliminated. This provided associative, not causal, evidence and mandates caution when interpreting these results.

## Conclusions

In summary, this meta-analysis mainly evaluated the role of NSE and S100B in predicting postoperative neurological complications in patients with AAAD. From the analysis, it was concluded that NSE and S100B were the ideal markers for diagnosing and S100B has a better predictive effect at 6 hours after the operation than 24 hours. In addition, NFL, IL-6, DD and coagulation factor R are expected to be biomarkers for postoperative neurological complications in patients with AAAD.

## Supporting information

**S1 Checklist. PRISMA checklist.**
(DOCX)

**S1 Table. Detailed risk assessment of publication bias.**
(DOCX)

**S1 Fig. Correction of NSE predicting neurological complications at 24 hours after operation.** (A) Funnel plot after subgroup analysis. (B) Analysis results after correction by shear

compensation method. (C) Funnel plot corrected by trim and filling method.
(TIF)

**S2 Fig. S100B predicting neurological complications at 6 hours after operation.** (A) Sensitivity analysis plot. (B) Funnel plot. (C) Analysis results after correction by trim and filling method. (D) Funnel plot corrected by trim and filling method.
(TIF)

**S3 Fig. Correction of S100B predicting neurological complications at 24 hours after operation.** (A) Sensitivity analysis plot before correction. (B) Funnel plot after sensitivity analysis. (C) Analysis results after correction by trim and filling method.(D) Funnel plot corrected by trim and filling method.
(TIF)

**S4 Fig. Correction of S100B predicting neurological complications at 6 and 24 hours after operation.** (A) Sensitivity analysis plot before correction. (B) Funnel plot after sensitivity analysis. (C) Analysis results after correction by trim and filling method. (D) Funnel plot corrected by trim and filling method.
(TIF)

## Author Contributions

**Data curation:** Dan Wu, Yifan Wang, Lingxiao Wang.

**Project administration:** Weixun Duan.

**Writing – original draft:** Yi Si.

**Writing – review & editing:** Weixun Duan, Jiangang Xie, Chujun Duan, Shanshou Liu, Qianmei Wang, Xiaojun Zhao, Junjie Li.

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
