## [Decision Letter · Decision Letter 0]

27 Jun 2022

PONE-D-22-13178Biomarkers for prediction of neurological complications after acute Stanford type A aortic dissection： a systematic review and meta-analysisPLOS ONE

Dear Dr. Li,

Thank you for submitting your manuscript to PLOS ONE. After careful consideration, we feel that it has merit but does not fully meet PLOS ONE’s publication criteria as it currently stands. Therefore, we invite you to submit a revised version of the manuscript that addresses the points raised during the review process.

We look forward to receiving your revised manuscript.

Kind regards,

Alessandro Leone, MD

Academic Editor

PLOS ONE

Journal Requirements:

Reviewers' comments:

Reviewer's Responses to Questions

**Comments to the Author**

1. Is the manuscript technically sound, and do the data support the conclusions?

Reviewer #1: Yes

Reviewer #2: Partly

Reviewer #3: Yes

2. Has the statistical analysis been performed appropriately and rigorously? 

Reviewer #1: Yes

Reviewer #2: N/A

Reviewer #3: Yes

3. Have the authors made all data underlying the findings in their manuscript fully available?

Reviewer #1: Yes

Reviewer #2: No

Reviewer #3: Yes

4. Is the manuscript presented in an intelligible fashion and written in standard English?

Reviewer #1: No

Reviewer #2: Yes

Reviewer #3: Yes

5. Review Comments to the Author

Reviewer #1: I have read with great interest regarding the biochemical prognostic markers of type 1 aortic dissection, which may help with management of this highly morbid condition. Below are my suggestions:

1) In the abstract, ". In addition, NFL, IL-6, DD and coagulation factor R are expected to be biomarkers for postoperative neurological complications in patients with AAAD." - the main text only mentions NSE and S100B as significant prognostic markers. I would revise or remove this phrase to avoid any confusions.

2) In the Introduction the authors mention: "The mortality of patients without surgical treatment is 90% in the first 24 hours and 50% in 48 hours". The cited article mentions overall mortality in medically managed AAAD to be around 50%, and it would be paradoxical to have 24hr mortality to be higher than 50%.

3) I would recommend expanding on the biologic role of NSE and S100B and its clinical significance on the Introduction rather than the Discussion.

4) Aside from above, the manuscript suffers from many small grammatical errors which sometimes hinder with legibility. I would recommend repeat proofreading to address these prior to publication.

Reviewer #2: Dear Authors, thank you for submitting your paper about this important topic.

I have the following questions:

- the majority of studies included in the paper come from China; to broaden the spectrum of disease, treatment and molecular assay, please include also the valuable contribution from other countries.

- In the plots experimental and control group are difficult for the reader to understand which group is; please change the labels accordingly

- some outputs are directly computer generated and are not easy to understand; please summarise the findings in one table.

Reviewer #3: Dear the authors of the manuscript entitled "Biomarkers for prediction of neurological complications after acute Stanford type A aortic dissection: a systematic review and meta-analysis"

Thank you for writing this systemic review and meta-analysis which highlighted an important subject in the field of aortic surgery

As youy explained, neurological complications are serious issues after aortic replacements and carry poor prognosis once happened

I can understand the limitations that were mentioned, but i do recommend the following points to be considered:

1. Please run an intense English language spell check through the whole manuscript

2. Please check the accuracy of the following written sentences in the introduction section:

A."The mortality of patients without surgical treatment is 90% in the first 24 hours and 50% in 48 hours". I guess the percentages need to reversed

B."Total aortic arch replacement is the first choice for saving patients' lives". I think the minority of the patients need total aortic arch replacements for Type AAAD

Thank you

6. PLOS authors have the option to publish the peer review history of their article (what does this mean?). If published, this will include your full peer review and any attached files.

Reviewer #1: No

Reviewer #2: No

Reviewer #3: **Yes: **Salah Eldien Altarabsheh

---

## [Author Response · Author response to Decision Letter 0]

17 Sep 2022

Dear Editor and Reviewers:

Thank you for your letter and for the reviewer’s comments concerning our manuscript entitled “Biomarkers for prediction of neurological complications after acute Stanford type A aortic dissection： a systematic review and meta-analysis”. Those comments are especially valuable and helpful for revising and improving our paper, as well as the important guiding significance to our researches. We have studied comments carefully and have made corrections which we hope meet with approval. Revised portion are marked in red in the paper. The main corrections in the paper and the responds to the reviewer’s comments are as following.

Reviewers' comments:

Reviewer #1: I have read with great interest regarding the biochemical prognostic markers of type 1 aortic dissection, which may help with management of this highly morbid condition. Below are my suggestions:

1) In the abstract, ". In addition, NFL, IL-6, DD and coagulation factor R are expected to be biomarkers for postoperative neurological complications in patients with AAAD." - the main text only mentions NSE and S100B as significant prognostic markers. I would revise or remove this phrase to avoid any confusions.

2) In the Introduction the authors mention: "The mortality of patients without surgical treatment is 90% in the first 24 hours and 50% in 48 hours". The cited article mentions overall mortality in medically managed AAAD to be around 50%, and it would be paradoxical to have 24hr mortality to be higher than 50%.

3) I would recommend expanding on the biologic role of NSE and S100B and its clinical significance on the Introduction rather than the Discussion.

4) Aside from above, the manuscript suffers from many small grammatical errors which sometimes hinder with legibility. I would recommend repeat proofreading to address these prior to publication.

Response:

Thank you for your professional, excellent and rigorous comments, they are very helpful for us and our manuscript. Your comments are sound as well as considerate. We have modified the manuscript according to your comments.

[1] Thank you for your professional and rigorous comments. We have revised the phrase of " In addition, NFL, IL-6, DD and coagulation factor R are expected to be biomarkers for postoperative neurological complications in patients with AAAD." to avoid any confusions. 

[2] Thank you for your professional comments. We have revised the sentecnes in the Introduction section, and would to express: "The mortality of patients without surgical treatment is 90% in the first 24 hours and overall mortality in medically managed AAAD to be around 50%". Please check and we apologize for this mistake.

[3] Thank you for your professional comments. In addition to the problems you mentioned we have checked and revised the Introduction and Discussion.

[4] Thanks for your careful comments. We have checked and revised the whole manuscript thoroughly.

If there is any problem, please don’t hesitate to contact me, may its blessings lead into a wonderful life for you and all whom you love!

Reviewer #2: Dear Authors, thank you for submitting your paper about this important topic.

I have the following questions:

- the majority of studies included in the paper come from China; to broaden the spectrum of disease, treatment and molecular assay, please include also the valuable contribution from other countries.

- In the plots experimental and control group are difficult for the reader to understand which group is; please change the labels accordingly

- some outputs are directly computer generated and are not easy to understand; please summarise the findings in one table.

Response:

Thanks for your professional, excellent and rigorous advice, we appreciate very much for your recognition of our study, we will humbly accept suggestions and make efforts to correct them. 

[1] Your comments are sound as well as considerate. We are sorry that we have not found more papers from other countries, this limitation was also mentioned at the end of the manuscript. Although related researches from other counties were limited, continuous attention to them is still going on. We hope more studies would be included in the future. Here, we apologize for this problem you mentioned.

[2] Thank you for your professional comments. We have changed the labels accordingly in Figs1-3 and S1-4Figs and added legends to make plots clearer.

[3] Thank you for your rigorous comments. We have summarized the findings in table3 accordingly, please check again. 

If there is any question, please don’t hesitate to contact me, may its blessings lead into a wonderful life for you and all whom you love!

Reviewer #3: Dear the authors of the manuscript entitled "Biomarkers for prediction of neurological complications after acute Stanford type A aortic dissection: a systematic review and meta-analysis"

Thank you for writing this systemic review and meta-analysis which highlighted an important subject in the field of aortic surgery

As you explained, neurological complications are serious issues after aortic replacements and carry poor prognosis once happened

I can understand the limitations that were mentioned, but i do recommend the following points to be considered:

1. Please run an intense English language spell check through the whole manuscript

2. Please check the accuracy of the following written sentences in the introduction section:

A."The mortality of patients without surgical treatment is 90% in the first 24 hours and 50% in 48 hours". I guess the percentages need to reversed

B."Total aortic arch replacement is the first choice for saving patients' lives". I think the minority of the patients need total aortic arch replacements for Type AAAD

Response:

Thanks for your professional, excellent and rigorous advice, we appreciate very much for your recognition of our study, thank you very much.

[1] Thanks so much for your positive comments. We have checked and revised the whole manuscript carefully.

[2] A Thank you for your professional comments. We have revised the sentence in the introduction now. New corrections have been marked in our manuscript named “Revised Manuscript with Track Changes”.

B Thank you for your rigorous comments. The pathogenesis, age and severity of aortic dissection in Chinese are obviously different from those in developed countries. The average age of onset in foreign patients is more than 60 years old, and most of them are aneurysmal dilatation caused by atherosclerotic lesions. Medical system in developed countries are relatively sound. Early detection, standardized systematic treatment and long-term follow-up observation of patients tend to leave fewer residual lesions, and the therapeutic effect is ideal. But in China, this age is about 46 years old, most of them are complicated with hypertension, and often cannot be treated in time, the disease is extensive and serious, and some patients need two or more operations. Therefore, total aortic arch replacement is the first choice for saving patients' lives in China and the details of the operation described in “Total Arch Replacement Combined With Stented Elephant Trunk Implantation”.

If there is any question, please don’t hesitate to contact me, may its blessings lead into a wonderful life for you and all whom you love!

Kindly regards

drjunjieli@126.com

---

## [Decision Letter · Decision Letter 1]

23 Jan 2023

Biomarkers for prediction of neurological complications after acute Stanford type A aortic dissection： a systematic review and meta-analysis

PONE-D-22-13178R1

Dear Dr. Li,

We’re pleased to inform you that your manuscript has been judged scientifically suitable for publication and will be formally accepted for publication once it meets all outstanding technical requirements.

Kind regards,

Eyüp Serhat Çalık

Academic Editor

PLOS ONE

Additional Editor Comments (optional):

Reviewers' comments:

Reviewer's Responses to Questions

**Comments to the Author**

1. If the authors have adequately addressed your comments raised in a previous round of review and you feel that this manuscript is now acceptable for publication, you may indicate that here to bypass the “Comments to the Author” section, enter your conflict of interest statement in the “Confidential to Editor” section, and submit your "Accept" recommendation.

Reviewer #3: All comments have been addressed

2. Is the manuscript technically sound, and do the data support the conclusions?

Reviewer #3: Yes

3. Has the statistical analysis been performed appropriately and rigorously? 

Reviewer #3: Yes

4. Have the authors made all data underlying the findings in their manuscript fully available?

Reviewer #3: Yes

5. Is the manuscript presented in an intelligible fashion and written in standard English?

Reviewer #3: Yes

6. Review Comments to the Author

Reviewer #3: Dear the authors

Thank you for taking in consideration the reviewers comments

I have no concerns about the manuscript

7. PLOS authors have the option to publish the peer review history of their article (what does this mean?). If published, this will include your full peer review and any attached files.

Reviewer #3: **Yes: **salah Eldien Altarabsheh

---

## [Editor Report · Acceptance letter]

26 Jan 2023

PONE-D-22-13178R1 

Biomarkers for prediction of neurological complications after acute Stanford type A aortic dissection：a systematic review and meta-analysis 

Dear Dr. Li:

I'm pleased to inform you that your manuscript has been deemed suitable for publication in PLOS ONE. Congratulations! Your manuscript is now with our production department. 

Kind regards, 

on behalf of

Dr. Eyüp Serhat Çalık 

Academic Editor

PLOS ONE